# Anomalous subtropical zonal winds drive decreases in southern Australian frontal rain

Acacia S. Pepler[1], Irina Rudeva[1]

[1]Australian Bureau of Meteorology, Melbourne, Australia

*Correspondence to*: Acacia Pepler (acacia.pepler@bom.gov.au)

**Abstract.** Cold fronts make a significant contribution to cool season rainfall in the extratropics and subtropics. In many regions of the southern hemisphere the amount of frontal rainfall has declined in recent decades, but there has been no change in frontal frequency. We show that for southeast Australia this contradiction cannot be explained by changes in frontal intensity or moisture at the latitudes of interest. Rather, declining frontal rainfall in southeast Australia is associated

with weakening of the subtropical westerlies in the mid troposphere, which is part of a hemispheric pattern of wind anomalies that modifies the extratropical zonal wave 3. Fronts that generate rainfall are associated with strong westerlies that penetrate well into the subtropics, and the observed decrease in frontal rainfall in southern Australia can be linked to a decrease in the frequency of fronts with strong westerlies at 25°S.

## 1 Introduction

Fronts are a major cause of rainfall and extremes in the global extratropics (Catto et al., 2012; Catto & Pfahl, 2013; Utsumi et al., 2017). While a large part of frontal precipitation is related to fronts embedded within extratropical cyclones (Dowdy and Catto, 2017), trailing cold fronts that are outside of the cyclone centre are an important cause of rainfall in many areas of the southern hemisphere midlatitudes and subtropics, particularly during the cool half of the year (Pepler et al., 2020; Utsumi et al., 2017).

In recent decades cool season frontal rainfall has decreased over parts of the southern hemisphere continents, including southwestern Australia (Risbey et al., 2013a), southeastern Australia (Pepler et al., 2021; Risbey et al., 2013b) and southern Africa (Burls et al., 2019). However, these studies typically found that this decrease in rainfall was not due to changes in the frequency of fronts. Indeed, studies have consistently observed little change in frontal frequency over the southern

hemisphere midlatitudes in reanalyses (Berry et al., 2011; Rudeva & Simmonds, 2015; Solman & Orlanski, 2014), although some decreases have been observed in the frequency of midlatitude cyclones (Pepler et al., 2021; Pepler, 2020b). Frontal frequency in southeastern Australia has had little change over this period, despite expectations of a southward shift in fronts due to observed  trends towards a positive Southern Annular mode (SAM) phase (Fogt & Marshall, 2020) and an intensification of the southern hemisphere storm track during winter (Chemke et al. 2022). Climate projections further

suggest a possible future increase in front frequency in southern hemisphere midlatitude regions (Blázquez & Solman, 2019; Catto et al., 2014), although in the subtropics frontal rainfall may still decline (Utsumi et al., 2016).

The observed decrease in frontal rainfall, in the absence of changes in frequency, suggests a change in either the moisture availability or the dynamics (e.g., intensity) of fronts that decreases the likelihood that they produce precipitation. Burls et al.

(2019) investigated this for South Africa and suggested that the decrease in frontal rainfall was related to increasing atmospheric pressure in the subtropics and Hadley cell expansion. Consistent with that, Sousa et al. (2018) showed a poleward migration of 'water corridors' due to an expansion of the semi-permanent South Atlantic high-pressure system followed by a displacement of the jet-stream during the latest drought in South Africa in 2015-2017. But the extent to which these results are transferrable to elsewhere in the southern hemisphere or to longer time periods is unclear.


Southeastern Australia (SEA) is an important agricultural region of Australia, as well as home to a large proportion of Australia's population. This region has experienced significant drying since the start of the Millennium Drought (1997 – 2009), particularly during the cool season May – October, which has been linked to an intensifying subtropical ridge and anthropogenic global warming (Timbal and Drosdowsky, 2013; Rauniyar and Power, 2020). The Millennium drought ended

in 2009, and was followed by heavy rain during the subsequent La Niña years 2010-2011. However, while average annual rainfall over the 2010-2018 period was close to the long-term average (Fu et al., 2021) this recovery is predominantly associated with increased rainfall during the warm season, when a lower proportion of rainfall is converted into streamflow. In contrast, rainfall during the hydrologically important cool months of the year remained below the long-term average during the post-drought period 2010-2019 (Bureau of Meteorology and CSIRO, 2020; DELWP, 2020). While much of the

decline in rainfall arises from decreases in both the frequency and intensity of rainfall from cyclones, there is an as-yet unexplained decline in the proportion of trailing cold fronts that produce rainfall (Pepler et al., 2021; Risbey et al., 2013b). In this study, we use front-centred composites to investigate the causes of declining frontal rainfall in southeastern Australia and the extent to which this can be linked to changes in frontal characteristics as well as large-scale circulation.

## 2 Data and methods

There are a large number of front identification methods, and front climatologies can be very sensitive to both the method chosen and the reanalysis product used (Soster and Parfitt, 2022; Schemm et al., 2015). While methods based on identifying a change in air mass via gradients of temperature or humidity are the most widely used, these can produce very high frequencies near coastlines (Thomas and Schultz, 2019; Schemm et al., 2015; Soster and Parfitt, 2022; Berry et al., 2011). They are also more sensitive to choices of reanalysis dataset and grid resolution than more complex front methods that also

incorporate wind information (Soster and Parfitt, 2022). While front methods that incorporate both wind and temperature information have shown improved skill at representing fronts (Bitsa et al., 2021; Biard and Kunkel, 2019), for southern

Australia a front detection method based solely on wind changes has shown good skill at detecting trailing cold fronts compared to manual fronts and particularly at detecting fronts associated with rainfall, with the majority of such fronts able to be confirmed by a temperature-based method (Hope et al., 2014; Pepler et al., 2020).


The wind-based front detection method is described in Rudeva & Simmonds (2015) and Simmonds et al. (2012). It compares two consecutive 6-hourly analyses of 10 m wind, and identifies a front when the horizontal wind shifts in direction from the northwest to southwest quadrant and the meridional wind increases by at least 2 m s$^{-1}$ over 6 h. Objective features are then identified, with the easternmost edge of the frontal region for the period t to t+6h identified as a front at time t, noting that

some studies suggest this may better approximate atmospheric fields at time t+6h (Papritz et al., 2014). This method is most useful for detecting cold fronts, which are the main fronts of relevance to rainfall in southern Australia. While tracking is performed for the whole southern hemisphere, for this study we require that fronts are at least 2 grid points long and have at least one point in southeast Australia (30-40°S, 135-150°E, Figure 1). Some supplemental analysis is also performed on fronts in southwestern Western Australia (SWWA; 23-38°S, 110-120°E), as this region is also experiencing a decline in cool

season frontal rainfall (Hope et al., 2006).

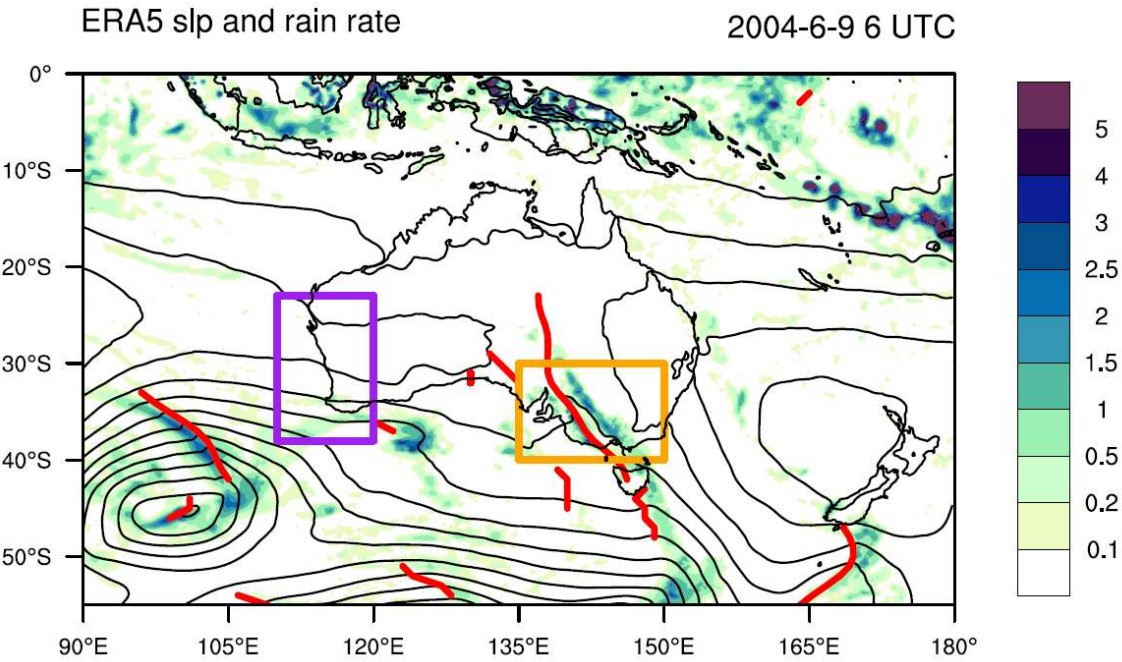

**Figure 1. The regions referred to in this study. Plot shows the ERA5 rain rate (shading, mm/h) and MSLP (contours) for a single time (600 UTC on 9 June 2004) in the Australian region, with identified cold fronts shown in red. Orange (SEA) and purple (SWWA) boxes show the regions used for identifying cold fronts relevant to this study.**


All data in this paper is obtained from the ERA5 reanalysis (Hersbach et al., 2020), with fronts identified every 6 hours on a 1° grid, noting that fronts identified on the native grid of high resolution show larger uncertainties in frequency between reanalyses (Soster and Parfitt, 2022). In addition to assessing the raw front tracks, ERA5 data is also used for front-centred composites and all other analyses in the paper. While reanalyses often evaluate poorly against observed rainfall measurements (Alexander et al., 2020), ERA5 generally evaluated well over Australia (Lavers et al., 2022). The predecessor to ERA5 (ERA-Interim) was found to generally perform well in simulating frontal rainfall and moderate rainfall intensities over the oceans near Australia (Lang et al., 2018), despite deficiencies in simulating pre-frontal and non-frontal rainfall, making ERA5 well suited to this study.

Geopotential height (Z), horizontal (u, v) and vertical (w) velocity, relative vorticity, and temperature (T) were extracted and analysed on eight vertical levels (1000, 925, 850, 700, 600, 500, 300 and 200hPa). Most results are presented for 700hPa where changes were most significant, but results were broadly consistent across a range of levels. Single-level variables included mean sea level pressure (MSLP), rain rate, total column water (TCW), 500-1000 hPa vertically integrated moisture flux including its zonal and meridional components (IVT; Reid et al., 2022), and the Phillips Criterion (PC), a measure of baroclinicity. PC was calculated as in Fahad et al. (2020):

$$PC = \frac{f^2(u_{500} - u_{lower})\Theta}{\beta g H(\theta_{500} - \theta_{850})}, \qquad\qquad\qquad \text{(Eq. 1)}$$

where H is the geometric height of the column—from the lower level (the average between 850 and 1000 hPa) to 500 hPa, and $\Theta$ is the reference potential temperature (300 K). While most of these variables are calculated instantaneously at the time of the front, we calculated the average rain rate from all hourly data between time t and t+6h; this can also be multiplied by 6 to represent the accumulated frontal rainfall over the corresponding period.

In some cases multiple fronts were identified in SEA at a single time, which could represent either two distinct fronts or a single system incorrectly broken into multiple parts by the tracking algorithm. To avoid double-counting any dates in composites, we created a single "merged front" for each timestep over the latitudes 20-50°S, which was used to extract front-centred data within 10° of longitude from the front location at each latitude for plotting and analysis. The merged front data was created using a four-step process, summarised in Figure 2:

1.  Identify all fronts that touch the region of interest (30-40°S, 135-150°E) at a given time (Figure 2a).
2.  If there is more than one front, first identify whether they overlap at any latitudes. If they do, iteratively remove the fronts with the smallest length within the region until there is only one front identified at each latitude (Figure 2b). Where multiple overlapping fronts have the same length, we prioritise retaining fronts that are nearer in longitude to the front with the most points in the region.

3. If there are still multiple fronts, check for cases where there is 3° or less of longitude difference between the neighbouring ends of the two fronts. If so, merge them into one front and set all missing points to the average of the longitudes at the end of each segment. Otherwise, remove the event with fewest points (Figure 2c).

4. Outside of the latitudes with an identified front, we infer an extended "front" longitude based on the last recorded front point, so that composites can be calculated over the full 20-50°S region (Figure 2d).

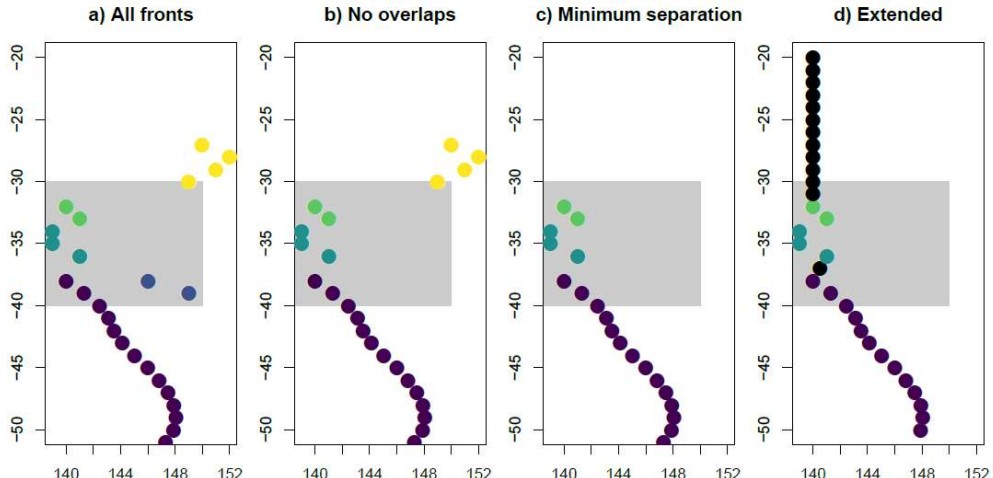

**Figure 2. a) An example of the front merger process for 0000UTC on 1992-12-27, with colours indicating the five distinct fronts identified at this time and a grey box indicating the SEA tracking domain. b) Of two overlapping fronts, the one with the largest number of points in the SEA region is retained, and the other is removed. c) A front at the far north of the region is removed, as it is too far east of the other identified front points. d) At latitudes with no identified front point, the longitude is inferred based on the closest frontal points (black dots).**

While fronts can be as short as one degree in length, and the total number of identified fronts is lower at 20°S than at 50°S (Rudeva and Simmonds, 2015), we calculate composites for latitudes between 20-50°S for all fronts regardless of their latitudinal position within that interval. This is because in many cases an identified cold front can interact strongly with weather systems to the north of the identified front extent, such as troughs and northwest cloudbands (Reid et al., 2019, 2022), and impact the atmospheric circulation and rainfall patterns well into the tropics (Narsey et al., 2017). Fronts are also often associated with atmospheric rivers that advect moisture from the tropics into higher latitudes (Reid et al., 2022), therefore understanding front-related circulation anomalies in the subtropics may help to understand changes in frontal rainfall.

Results are presented for the cool season May-October, with the twenty-year periods 1980-1999 and 2000-2019 compared and statistical significance calculated using a Student's t-test. We additionally use three periods, 1979-1996, 1997-2009 and 2010-2019 in some instances to test for recovery in frontal rainfall following the Millennium drought. Unless otherwise specified, front-centred averages are calculated within +-5° of the front central longitude, representing the region with the

majority of frontal rainfall, with a focus on southeast Australian latitudes 33-38°S. As the anomalous trends in rainfall are in particular a feature of trailing fronts (Pepler et al., 2021), we used a dataset of Australian cyclones detected using ERA5 (Pepler, 2020a) to compare changes in trailing fronts with those for fronts embedded in extratropical cyclones. Noting that in earlier studies the area of cyclone rainfall is often taken as 10-12° around the cyclone centre (Hawcroft et al., 2012; Pepler et al., 2020), we considered a front to be embedded in a cyclone if a cyclone centre was detected within the 135-150°E, 20-45°S region (Hawcroft et al., 2012; Pepler et al., 2020), or a trailing front if there was no cyclone in this region. During May-October, 70% of detected fronts are considered trailing fronts, as are 55% of fronts with rain rates exceeding 0.1 mm/h in southeast Australia, as rain rates are higher in cases where cyclone and front areas overlap (Dowdy & Catto, 2017; Pepler et al., 2020).

Pearson's correlation coefficients are calculated using linearly detrended data to assess relationships between seasonal mean frontal characteristics and the intensity (STRI) and position (STRP) of the subtropical ridge calculated over 140-150°E (Timbal and Drosdowsky, 2013). We also calculate correlations with the Troup (1965) Southern Oscillation Index (SOI; http://www.bom.gov.au/climate/enso/soi/), an indicator of the El Niño-Southern Oscillation (ENSO); the Dipole Mode Index (DMI), an indicator of the Indian Ocean Dipole (IOD: Saji et al., 1999; https://stateoftheocean.osmc.noaa.gov/sur/ind/dmi.php); and the Southern Annular Mode (SAM: https://www.cpc.ncep.noaa.gov/products/precip/CWlink/daily_ao_index/aao/aao.shtml). Statistical significance is assessed using a Student's t-test for $p < 0.05$; this is calculated using seasonal mean data for differences between periods, and between all fronts when identifying significant differences in structure between the wet and dry subsets.

## 3 Changes in frontal rainfall

We first assess changes in front statistics using the raw ERA5 output over 1979-2019. There is a front detected somewhere in southeast Australia at approximately 50% of time steps during May-October, with no significant difference in frequencies between the Millennium drought in 1997-2009 (49%) and the non-drought periods (50.6%). No change in front frequency is found when comparing the periods 1980-1999 and 2000-2019 for southeast Australia as a whole, or for the frequency of fronts detected for each latitude band within this region (Figure 3b). There is also no change in the average front intensity at these latitudes, defined as the strength of the change in meridional winds. There is a weak decrease in front frequency at the equatorward edge of the region influenced by fronts (20-25°S), from 22.3% to 21.5% of hours per season, but this is not statistically significant.

Figure 3a shows the average rain rate for hours with an identified front in southeast Australia, centred on the longitude of the front. Rain rates greater than 0.1 mm/h are recorded within 5 degrees of longitude on either side of the front location, with heaviest rain rates slightly west of the front line. We thus define frontal rainfall to be the average over a 10° region centred

 on the frontal line (0°), and use the 0° line to separate rainfall into "prefrontal" and "postfrontal" rainfall. The likelihood of frontal rainfall and the average annual frontal rainfall are highest south of 37°S. While the frequency of detected fronts remains higher than 100/season as far north as 25°S, the average rainfall from fronts decreases rapidly north of 33°S, suggesting that the northern edge of fronts is typically too weak or dry to produce rainfall.

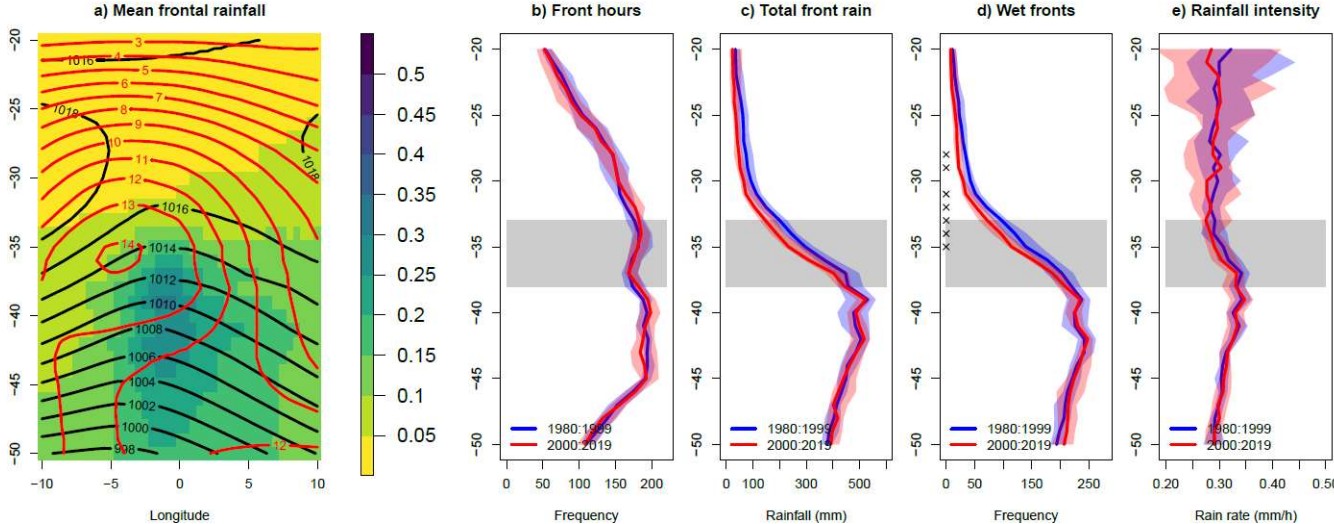

**Figure 3. a) Average rain rate (shading, mm/h), MSLP (black contours) and 700 hPa zonal wind (red contours) for all hours with at least one front in SEA during May to October, centred on the longitude of the front. b) Average number of timesteps in May-October with at least one front identified at each latitude within the longitudes 135-150°E, based on the full dataset of southern hemisphere fronts prior to the merging process. Blue and red lines show the median for 1980-1999 and 2000-2019, respectively, while shading shows the interquartile range from seasonal data. c-e) Total accumulated frontal rainfall (c), number of fronts with rainfall > 0.1 mm/h (d) and average rain rate where rain is >0.1 mm/h (e) across all May-October timesteps with a front detected in SEA using the merged front dataset in 1980-1999 and 2000-2019. In b-e, crosses indicate the differences are statistically significant at p<0.05.**

Comparing the periods 1980-1999 and 2000-2019, total May-October frontal rainfall has declined at all latitudes north of 38°S (Figure 3c). In SEA (33-38°S) there has been a statistically significant decrease in the mean rain rate across all front days, from 0.155 mm/h to 0.138 mm/h (-11%). This accumulates to a 32mm (9.4%) decline in total frontal rainfall per season between the two periods, with larger relative declines in rainfall further north where average frontal rainfall is smaller. While this change is not statistically significant (p=0.14), it is similar in magnitude to the annual rainfall decline during the Millennium Drought (Van Dijk et al., 2013) and to declines in frontal rainfall reported in previous studies (Risbey et al., 2013b; Pepler et al., 2021).

There was a statistically significant decline in frontal rainfall between 1979-1996 and 1997-2009 (-14%, p=0.03), with a partial recovery during 2010-2019 (9% below the 1979-1996 period). The 2010-2019 period had very high variability, with very high frontal rainfall totals during 2010 and 2016 but low totals in other years, which explains the reduced significance of trends calculated using the later period. Recovery during the later period occurred mostly in the early season, April-June, with the frontal rainfall anomaly in July-October during 2010-2019 (-15% compared to July-October 1979-1996) similar to that in 1997-2009 (-17%).

This decline is not related to any change in front frequency (Figure 3b) but due to a decrease in the average rainfall intensity calculated across all fronts. This decrease in intensity is due to a decrease in the likelihood that a front will produce measurable rain in these latitudes (Figure 3d) and a corresponding increase in dry fronts, consistent with Pepler et al. (2021); for fronts that produce rainfall, the average rainfall intensity has not changed (Figure 3e). The number of fronts per season with rainrates exceeding 0.1 mm/h in SEA decreased from 164 to 140 between 1979-1996 and 1997-2009 (-14%, p=0.01), with little recovery over the recent period 2010-2019 (145 per season). There is a decrease in the likelihood that a front will produce rainfall whether or not the front is collocated with a cyclone. The total accumulated rainfall in the prefrontal region (0 to +5°) declines by 14%, which is larger than the rainfall decline in the postfrontal region (-5%). This is in contrast to Burls et al. (2019), who found that the largest decline in South African rainfall occurred on postfrontal days.

## 4 Comparison of wet and dry fronts

Given the observed decrease in the proportion of fronts that generate rainfall, we now investigate how the characteristics of wet and dry fronts differ during 1980-1999. We define a front as being wet if the average rainrate over 33-38°S and within ±5° of the frontal line is at least 0.1 mm/h, which is satisfied by 51% of May-October fronts in 1980-1999.

Wet and dry fronts differ in many aspects (Figure 4a-f). Wet fronts have more negative (stronger) vertical velocities at all levels, with the largest differences in the mid troposphere (500-700hPa); stronger cross-front gradients in temperature as well as meridional wind (not shown); and stronger relative vorticity, particularly behind the front. These variables all indicate that rain-bearing fronts are stronger in southeast Australia than those that produce little rain. Total column water and integrated vapour transport (not shown) are also higher for rain-bearing fronts at SEA latitudes as well as to the north, noting that the moisture for rain events in southeastern Australia is generally sourced from the oceans to the south (Holgate et al., 2020). Rain-bearing fronts show a higher pre-frontal PC, a measure of baroclinicity, to the north of 35°S, and lower prefrontal baroclinicity further south. This decrease in the PC to the south may be explained by consumption of baroclinity during rainfall, which leads to even stronger reduction in PC after the front. On the other hand, higher PC in rain-bearing fronts to the north of 35°S promotes stronger moisture uplift.

While there is only a small difference in mean zonal winds at the latitudes of interest, in fronts that produce rainfall there are stronger westerlies north of 33°S but weaker zonal winds south of 38°S (Figure 4f). This reflects a change in the mean patterns of winds at all levels, with the strongest relationships between rainfall and zonal winds found at 700hPa. For instance, the latitude where 700hPa zonal winds are strongest during wet fronts is 34.9°S, 6° further north than for dry fronts (40.9°S). This means that wet fronts have strong 700hPa zonal winds (defined as u > 10 m/s) reaching 25.4°S on average, and average zonal wind speeds of 10.8 m/s at 23-27°S. In contrast, strong westerlies only reach 30.6°S for dry fronts, and wind speeds near 25°S are half as strong (4.7 m/s). There is also a smaller but statistically significant difference in the northernmost latitude where any front is identified in the longitudes 135-150°E, which is 26.5°S for fronts with rainfall and 27.9°S for dry fronts.

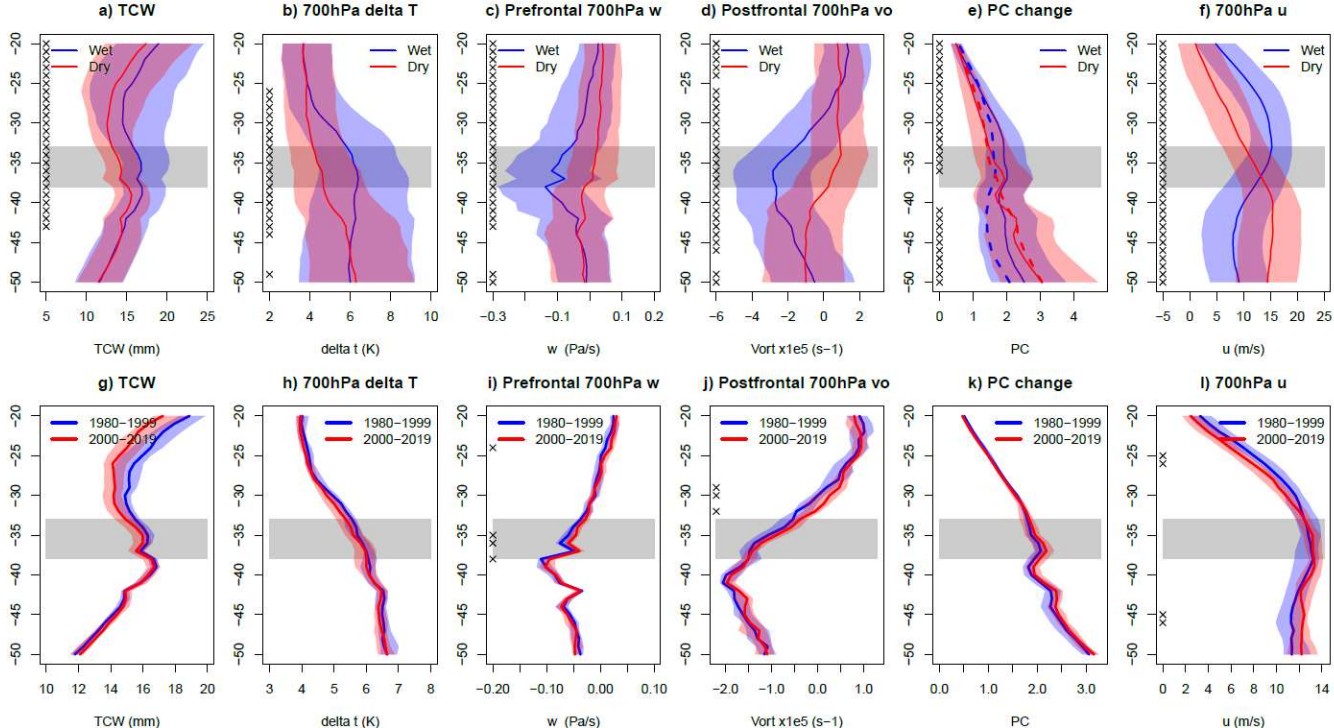

**Figure 4.** (a-f) Mean (line) and interquartile range (shading) of 6 variables at each latitude for May-October fronts in 1980-1999 with rain rates of at least 0.1 mm/h at 33-38°S compared to dry fronts: a) Total column water (±5° of front); b) Difference between maximum and minimum 700hPa temperature (±10° of front), indicating strength of the temperature change; c) Prefrontal 700 hPa vertical velocity (0° to +5°); d) Postfrontal (-5 to 0°) 700hPa relative vorticity; e) Mean prefrontal Phillips Criterion (0 to +5°), a measure of baroclinity, with dashed lines showing the postfrontal medians (-10° to -5° behind front); f) 700hPa zonal wind (±5° of front); (g-l) As in top row, but for the median and interquartile range of the seasonal mean value across all fronts in 2000-2019 vs 1980-1999. Crosses indicate where the two sets are statistically significantly different for p<0.05 at that latitude.

## 5 Changes in frontal characteristics

Having identified key aspects of fronts that differ between wet and dry fronts, we now investigate how these have changed between 1980-1999 and 2000-2019, to help identify any changes in frontal mean characteristics that decrease the likelihood of frontal rainfall.

Comparing the average across all fronts in 1980-1999 with fronts in 2000-2019 (Figure 4g-l), there has been a very small decline (-1.3%) in the average TCW at 33-38°S on front days, suggesting moisture availability is unlikely to be a major contributor to changes in frontal rainfall. Changes in metrics of front intensity at these latitudes such as the average change in meridional wind speed or change in temperature across the frontline are also very small (<3%). There is no statistically significant change in baroclinicity measured by the Phillips Criterion in southeast Australian latitudes, although there is a reduction to the south. There is a statistically significant decrease (-14%) in the prefrontal 700hPa vertical velocity in SEA, coinciding with the latitudes of the largest rainfall decline (Figure 4c, i), and a weak, non-significant increase in postfrontal vertical velocity. There is also a statistically significant weakening of post-frontal relative vorticity over 33-38°S (-11%), with larger changes in vorticity over 28-32°S. While these changes are generally weak, and not necessarily aligned with the latitudes most relevant to our region of interest, together they are suggestive of an overall weakening of uplift in the frontal area.

There are larger changes in both moisture variables and indicators of frontal intensity between the two periods at latitudes north of 30°S, including mean meridional wind change (not shown), relative vorticity, and mean zonal winds. This indicates a weakening of the northward edge of the front and a southward shift of frontal features. Between 1980-1999 and 2000-2019 there has been no change in the average northernmost latitude with a detected front. However, the mean northernmost latitude of zonal winds exceeding 10 m/s has shifted from 28.0°S to 28.8°S, associated with a 12% decline in the mean zonal wind speed at 23-27°S, noting that these two variables are strongly correlated (r=+0.74). Associated with the weakening zonal winds, there has also been a reduction in mean IVT at 28-33°S (not shown), indicating a reduction of moisture flux at the northern edge of the front, although there has been no change in IVT over SEA (33-38°S).

To assess the extent to which various front characteristics can be tied to changes in frontal rainfall we apply multiple linear regressions between frontal rainfall and one or more explanatory factors over 1980-1999. We then apply the regression coefficients to the 2000-2019 period to calculate the reduction in mean rainfall expected from observed changes in the predictors, and divide this by the observed rainfall change between the two periods to calculate the proportion of rainfall change explained by those factors. We found that a single linear regression between mean zonal winds at 23-27°S and mean frontal rainfall at 33-38°S over 1980-1999 is able to explain 59% of the decrease in average frontal rainfall. The proportion of rainfall decline explained can be increased by adding either prefrontal vertical velocity (71%) or TCW (81%); as TCW

and w are correlated (-0.42), there is no additional predictive value from a three-variable regression. Predictive skill is similar but slightly lower if we use the northernmost latitude of zonal winds exceeding 10 m/s combined with TCW (71%). In contrast, single linear regressions with either vertical velocity (32%) or TCW (25%) explain only a small proportion of the rainfall change.

These multiple linear regressions typically underestimate the higher end of frontal rainfall, as the relationship between frontal rainfall and both zonal winds and TCW are nonlinear. Fronts with strong westerlies extending to at least 23°S have average rain rates over 33-38°S more than three times higher than fronts where strong westerlies are only observed up to 30°S. During 1980-1999, only 27% of fronts had westerlies extending to 23°S, but these fronts explained 42% of all frontal rainfall over SEA.

Between 1979-1996 and 1997-2009, i.e., before and during the Millennium Drought, there was a 28% decrease in the frequency of fronts with 700hPa westerlies extending north of 23°S, and a 34% (54 mm, p=0.01) decrease in their accumulated rainfall over SEA (Figure 5). There has been little recovery following the Millennium drought, with frequencies in 2010-2019 25% below the 1979-1996 average and rainfall 25% below the 1979-1996 average. As there is no overall change in the frequency of fronts during or after the drought, the decrease in the number of fronts with westerlies north of 23°S is balanced by an increase in the frequency of fronts where westerlies are poleward of 23°S, and a corresponding weak increase in associated rainfall (+8 to +13mm). The decrease in the frequency of fronts where westerlies extend northward of 23°S is thus sufficient to explain the entirety of the observed change in frontal rainfall at 33-38°S during the period since 1997.

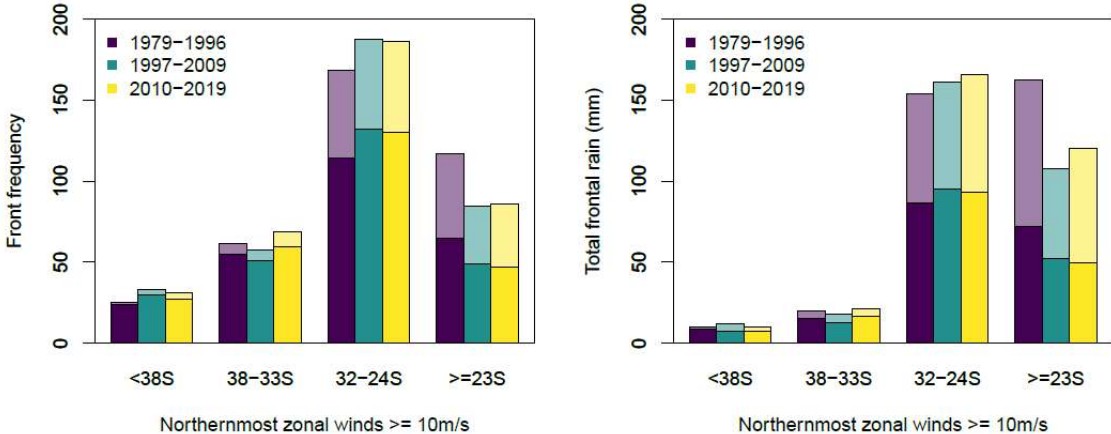

Figure 5. The average frequency (left) and total accumulated rainfall in 33-38°S (right) for May-October fronts in 1979-1996, 1997-2009 and 2010-2019, separated by the northernmost latitude where 700hPa zonal winds exceed 10 m/s. Each bar shows the total across all fronts, with darker shading indicating the component from trailing fronts and lighter shading showing the component from fronts embedded in a cyclone.

To test the role of cyclones in this result, we further separated our analysis between trailing fronts and fronts associated with a cyclone. Almost half (44%) of fronts with westerly winds extending north of 23°S had an associated low, which is higher than the frequency of lows across all fronts (29%). When comparing 1979-1996 and 1997-2009, there was a 27% decline (p=0.03) in rainfall from trailing fronts and a larger 39% decline (p=0.008) in rainfall from fronts that *co-occurred* with a cyclone, indicating that declines in frontal rainfall can in part be linked to the observed decrease in the frequency of low pressure systems in this period (Pepler et al. 2021), as any change in low pressure systems will also affect rainfall from embedded fronts. In comparison, the 2010-2019 period has seen a partial recovery of rainfall from embedded fronts (22% below the 1979-1996 average), but no recovery in rain from trailing fronts (-31%). This may indicate the change in rainfall from trailing fronts is playing an increasingly large role in overall rainfall declines, although differences between 1997-2009 and 2010-2019 are not statistically significant. Meanwhile, for fronts where strong westerly winds are south of 23°S, both those with and without cyclones have increased in frequency, indicating that the westerly winds are playing a stronger role in rainfall changes than interactions with cyclones.

## 6 Links to large-scale circulation

While we have demonstrated that declines in rainfall at 33-38°S can be explained by a weakening and southward shift of the northern edge of the front, this raises a subsequent question: given that we have shown no decrease in the frequency or intensity of fronts such as their 700hPa longitudinal temperature gradient, measures of frontal baroclinicity such as the Phillips criterion, or even the latitude of fronts as identified using a wind-based front identification scheme, what is the driver of this weakening? We propose that this is related to changes in the atmopheric extratropical circulation in the SH.

Between 1980-1999 and 2000-2019 there has been a change in MSLP and wind anomalies suggestive of wave number 3 (Figure 6a). The strongest anomaly in the MSLP is observed over the southern Atlantic ocean, associated with an easterly anomaly in both 700hPa and 300hPa winds around 40°S and westerly anomaly around 60°S. Weaker trends towards higher pressure and easterly wind anomalies are also evident around 100°E and 220°E, with corresponding westerly anomalies to the south. Areas of anticyclonic anomalies are interspersed with cyclonic anomalies, which are typically slightly poleward. Additional analysis of the meridional wind showed that these changes act to modify the climatological zonal wave 3 (ZW3), defined as the leading EOF of the monthly meridional wind as in Goyal et al. (2022) but using 700 hPa where we see the largest wind change in fronts (Figure S1). However, changes to the ZW3 do not represent a perfect zonal wave, possibly due to interactions with other wave numbers and local forcings, hence, no trend in either intensity or location of the climatological ZW3 was found. With the exception of very strong anomalies such as in the southern Atlantic and most northward parts of the two other anticyclonic anomalies, most of other changes in the SH circulation are not statistically significant, including in eastern Australia (Figure S2).

This hemispheric wavy pattern of anomalies is still apparent when data is separated into days with (Figure 6b) and without (Figure 6c) a front in southeast Australia. However, on front days the wave pattern is amplified and there are easterly wind anomalies at 700hPa over northeastern Australia, with weak easterly anomalies extending to 300hPa, and westerly anomalies

as well as below average pressure to the south of Australia, consistent with the composites shown in Figure 4. (We note here that the low pressure anomaly to the south of Australia is located further south than the area that was considered for separation between embedded and trailing fronts). The topography of New Zealand then acts as a barrier to the intensified westerlies around 45°S, with return flow potentially contributing to the anticyclonic anomaly in the Tasman Sea and therefore the stronger subtropical easterly anomalies. This pattern of zonal wind anomalies is weaker or absent on days with

no front in SEA, with increasing MSLP over the Australian Bight and westerly anomalies at 300hPa over northeast Australia. This demonstrates a more complex relationship between changes in the mean state and synoptic anomalies. An anticyclonic anomaly to the east of Australia on front days suggests Rossby wave breaking, noting that SEA is known for high number of Rossby wave breaking events (de Vries, 2021) and cut-off lows (Portmann et al., 2021).

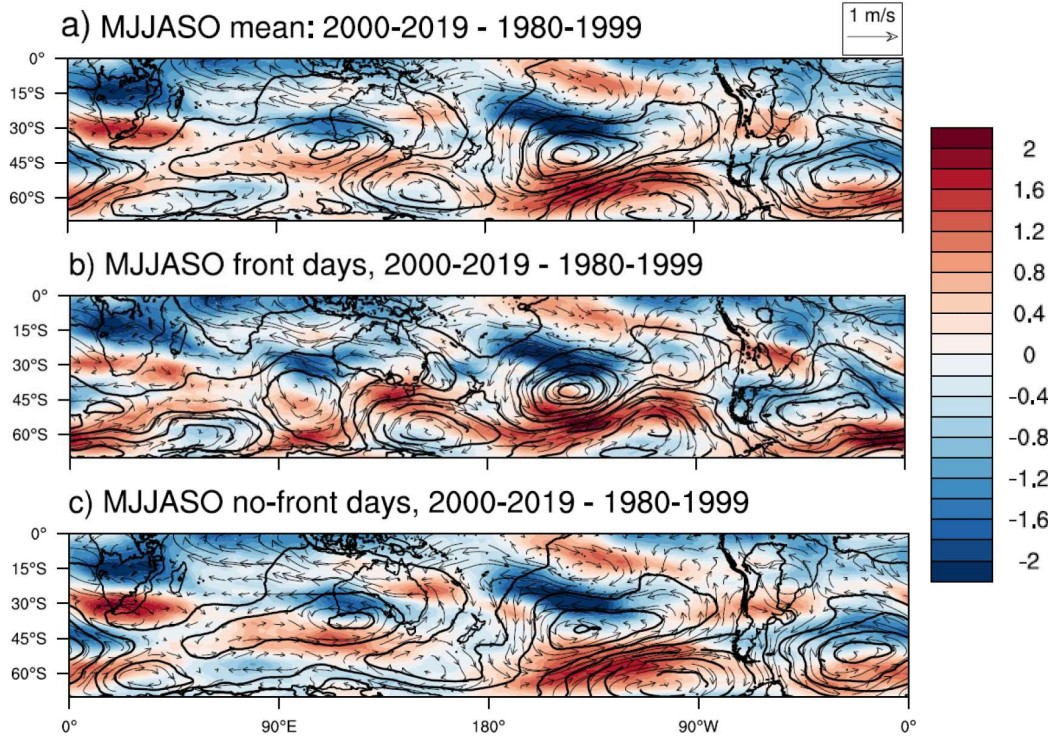

**Figure 6. Anomalies of May-October 300hPa zonal wind speed (shading, every 0.2 m/s), 6-hourly mean sea level pressure (black contours, every 0.5 hPa), and 700hPa winds (vectors) between 1980-1999 and 2000-2019 in ERA5 for a) Front hours only, b) Hours without a front in southeast Australia, and c) Seasonal mean change.**

To test the statistical significance of wind changes, we calculated the average zonal wind anomalies for each latitude band over eastern Australia, 135-150°E. There are no statistically significant changes in seasonal mean 300hPa winds between 1980-1999 and 2000-2019 at any latitude, with weak westerly anomalies (0.3 m/s) around 25°S, easterly anomalies (~ -0.35 m/s) around 30-35°S, and stronger westerly anomalies southward (Figure 6a). This mean pattern combines very different patterns over eastern Australia between days with and without fronts, which had trends of opposite signs over most of 20-40°S. In contrast to the mean change, there was a weak subtropical easterly anomaly on front days, averaging -0.4 m/s over 20-25°S, in agreement with an anticyclonic anomaly to the east of Australia. Subtropical anomalies are more consistent at 700hPa with a mean zonal wind change of -0.6 m/s at 20-25°S, which is stronger for front days (-0.8 m/s, p=0.08) than for no-front days (-0.4 m/s).

## 7 Drivers of interannual variability

To test the role of major climate drivers in observed changes, we calculated the detrended correlations between the annual number of fronts in May-October, as well as associated winds and rainfall with a number of climate indices over the same months (Table 1). The overall frequency of fronts is negatively correlated with the intensity and position of the subtropical ridge, with fewer fronts when the ridge is strong or shifted to the south, noting that cold fronts are typically linked to the strom tracks and westerlies and are less likely to be identified in easterly wind regimes to the north of the ridge. These correlations are a result of very strong negative correlations between both indices (STRI and STRP) and the frequency of wet fronts, with weaker positive correlations between the STR and dry fronts. The STR is also very strongly correlated with the average northernmost latitude of strong westerlies for fronts and the number of fronts with strong westerlies north of 23°S. However, it is only weakly correlated with the seasonal mean zonal winds at 20-25°S, with stronger negative correlations between STRI and zonal winds between 26-35°S. This may indicate a role of the STR in keeping frontal weserlies south and thus influencing frontal rainfall; however, the STR itself is also influenced by pressure variations associated with synoptic systems so the relationship may be more complex.

Table 1. Detrended Pearson's correlations between front frequency, rainfall, and the northernmost latitude with 700 hPa zonal wind >= 10 m/s (strong winds) and 6 climate indices for May-October, 1979-2019 (1982-2019 for DMI and NINO3.4). Also shown are the partial correlations for NINO3.4 and DMI. Bold indicates significance for p<0.05.

| Variable | STRI | STRP | SOI | NINO3.4 | DMI | SAM | NINO3.4$_{DMI}$ | DMI$_{NINO3.4}$ |
|---|---|---|---|---|---|---|---|---|
| Front days | **-0.53** | **-0.58** | 0.28 | -0.19 | -0.19 | -0.12 | -0.13 | -0.09 |
| Total front rainfall | **-0.66** | **-0.53** | **0.56** | **-0.40** | **-0.68** | 0.08 | -0.11 | **-0.62** |
| Number of wet fronts | **-0.76** | **-0.65** | **0.56** | **-0.42** | **-0.60** | -0.01 | -0.17 | **-0.53** |
| Number of dry fronts | **0.34** | 0.18 | **-0.35** | 0.28 | **0.48** | -0.09 | 0.04 | **0.45** |

| | | | | | | | | |
|---|---|---|---|---|---|---|---|---|
| Average northmost latitude of U>=10m/s | **-0.78** | **-0.65** | **0.57** | **-0.36** | **-0.67** | -0.08 | -0.04 | **-0.63** |
| Number of fronts with strong winds north of 23S | **-0.73** | **-0.60** | **0.51** | -0.30 | **-0.65** | 0.02 | 0.03 | **-0.62** |
| Number of fronts with strong winds south of 23S | **0.38** | 0.22 | **-0.34** | 0.17 | **0.55** | -0.10 | -0.13 | **0.56** |

There were no statistically significant correlations between the seasonal mean SAM and front frequency or rainfall, which may be a consequence of the large intraseasonal variability in SAM. There was also no correlation between fronts and NINO3.4 after accounting for covariations with DMI, with the stronger correlations with SOI potentially due to interactions between the Indian Ocean Dipole and sea level pressure at Darwin. Interestingly, however, while DMI was not strongly correlated with the seasonal frequency of fronts, it had a very strong correlation with both total frontal rainfall and the frequency of wet fronts, which was independent of any effect of ENSO. Notably, the three seasons with the highest total frontal rainfall during the period all occurred during negative IOD conditions (1992, 2010 and 2016), and the three seasons with lowest frontal rainfall occurred under positive IOD (1982, 1994 and 2006).

While the link between IOD and frontal rainfall could in part be associated with increases in moisture from the tropics, the IOD was also associated with a shift in the average northernmost latitude of strong frontal westerlies, consistent with the extratropical pathway of IOD impacts (Cai et al., 2011) and the link between the IOD and the mean westerlies in southeast Australia (Pepler et al., 2014). Given that the IOD is most active in spring, the influence of IOD on the Australian rainfall is strongest during spring (e.g., Cai et al. 2011). The correlation between the DMI and total frontal rainfall also strengthens from -0.39 during May-July to -0.65 during August-October. In both seasons negative IOD is associated with an increased frequency of wet fronts and a northward shift of frontal westerlies, but no significant change in the total number of fronts.

These results show that wet and dry fronts can have very different relationships with climate drivers, resulting in the weaker relationships that have previously been identified between front frequency and DMI (Rudeva and Simmons 2015). Negative Indian Ocean Dipole events and a weaker subtropical ridge (negative STRI) allow front-related westerlies to extend further north, resulting in an increased frequency of wet fronts, although statistically significant correlations between these drivers and the seasonal mean zonal wind are only identified south of 26°S. Note that the DMI is correlated with both the intensity (+0.58) and position (+0.32) of the subtropical ridge during this season, so these relationships are not independent, and an equatorward shift in fronts may itself cause changes in the pressure fields which are used to calculate the STR.

During the period 1979-2019, there has been a weak increasing trend in the intensity of the sutropical ridge, of 0.26 hPa/decade (p=0.09), consistent with longer-term increases over the historical record (Timbal & Drosdowsky 2013). This may partially explain the observed trends in wet fronts: while there is a statistically significant linear trend in the number of wet fronts over 1979-2019 (-0.92 fronts/y, p=0.03), this trend becomes weaker and nonsignificant after removing variability associated with the subtropical ridge intensity (-0.35 fronts/y). Similarly, while there was a -0.4°/decade linear trend in the average northernmost latitude with U700>=10m/s (p=0.01), removing variability associated with STRI decreased this trend to -0.2°/decade (p=0.06).

## 8 Changes in fronts in southwest western Australia

The very strong easterly anomalies at 700hPa and 300hPa identified over southwest Western Australia in Figure 6 raise the question of whether changes in the background zonal winds are also playing a role in rainfall declines in this region. Similar to southeast Australia, SWWA (110-120°W) has seen little decrease in the frequency of fronts between 1980-1999 and 2000-2019, but a statistically significant 11% (38mm) decrease in total frontal rainfall, particularly prefrontal rainfall (-21%), which is linked to a decrease in the frequency of rainbearing fronts (Figure S3). The average differences between wet and dry fronts are also similar between the two regions, with the most significant change relevant to frontal rainfall observed in the zonal winds to the north of SWWA (Figure S4). And, consistent with SEA, there has been a 21% decrease in the frequency of fronts in SWWA with strong 700hPa westerlies extending north of 23°S and a 44mm (-23%) decline in associated rainfall, which explains the entirety of the observed decline in frontal rainfall. The extratropical circulation on SWWA front days resembles the zonal wave 3 pattern shown for SEA in Figure 6b, with the low pressure anomaly to the south of Australia shifted westward (Figure S5). While most of the decline in frontal rainfall in SWWA can also be attributed to fronts with westerlies reaching at least 23°S, the decline over time is more linear than SEA, with lower frontal rainfall during 2010-2019 than 1997-2009 (Figure S6).

## 9 Discussion and Conclusions

Frontal rainfall is influenced by a large number of factors, including available moisture (e.g., TCW) and frontal dynamics (e.g., temperature/wind change, vertical velocity, and relative vorticity). This is consistent with other studies (Solari et al., 2022) and suggests that changes in any of these variables could result in changes in frontal rainfall. However, while the frequency of rain-bearing fronts is decreasing during the cool season in many parts of the southern hemisphere midlatitudes, including in southeast Australia, this decrease is occurring despite little change in front frequency (Burls et al., 2019; Pepler et al., 2021; Risbey et al., 2013b), as well as little change in metrics of frontal moisture or traditional measures of front intensity, beyond a slight weakening of average prefrontal vertical velocity and postfrontal vorticity.

There has been no observed change in the mean latitude of tracked fronts in southeast Australia during the cold season, consistent with previous studies (e.g., Rudeva & Simmonds 2015), despite decreases in the frequency of associated cyclones. However, if the northernmost edge of the front is instead defined as the northernmost latitude with strong westerly winds, here defined as 700hPa zonal winds >= 10 m/s, we see a 0.8° southward shift in the mean equatorward extent of fronts over this period. We find a particularly strong decline in the frequency of fronts with strong westerlies north of 23°S, which were responsible for 42% of frontal rainfall at 33-38°S between 1980-1999. This highlights an aspect of frontal intensity and associated rainfall that is worthy of further research, and may be relevant to other regions of the southern hemisphere. There has also been a decrease in the overall intensity of the subtropical westerlies over eastern Australia and many other areas of the southern hemisphere, consistent with Simmons (2022), although the trend in the mean state is weaker than on front days.

Frontal winds and rainfall have shown little recovery since the end of the Millenium Drought, and may be part of a longer-term decline in cool season rainfall over this region. While the total decrease in rainfall from fronts is relatively small, a 9.4% decline in frontal rainfall from 1980-1999 to 2000-2019 in southeast Australia, even small changes in rainfall can be magnified to much larger changes in streamflow and corresponding hydrological impacts. This was seen during the Millenium Drought, when a 11% decline in total annual rainfall was magnified to a 46% decline in streamflow in southeast Australia (Van Dijk et al., 2013). Changes in frontal rainfall may potentially play an important role in these declines, as fronts contribute strongly to the number of days with low to moderate rainfall intensity (Pepler et al., 2020) and the number of rain days is an important predictor of streamflow changes in southeast Australia (Fu et al., 2021). Consequently, any continuation of declines in frontal rainfall over coming decades may be of critical importance in catchments where runoff has yet to recover after the Millenium Drought (Peterson et al., 2021).

The link identified in this paper between subtropical wind changes and changes in frontal rainfall is a statistical link, and would require further modelling studies to understand the dynamical interaction between subtropical winds and rainfall ~10° further south. However, there are a number of potential mechanisms. The southward shift in frontal westerlies could simply be an indication of a southward shift in the latitude at which fronts occur, which would suggest that the front detection method based on wind changes is deficient at detecting the equatorward extent of cold fronts and subtle changes at these latitudes. However, front detection methods based on other variables such as temperature gradients also do not identify any changes in frontal frequency (Pepler et al., 2021; Berry et al., 2011). Another potential mechanism is a link between weakening subtropical westerlies and a reduction in atmospheric moisture flux into the region; while there has been little change in total column water or IVT in SEA, both TCW and IVT have decreased in the region 28-33°S associated with the weakening subtropical winds. Finally, while we found only a weak change in vertical velocity and no significant change in PC, changes in other factors such as convective inhibition or the level of condensation in a warming climate could also reduce the ability of fronts to generate rainfall, particularly during the cool season and moderate rain events where convection is weak (Rasmussen et al., 2020).

Timbal & Drosdowsky (2013) identified an intensification of the subtropical ridge during the early 21st century, which they proposed as a major cause of rainfall declines in southeast Australia during the years 1997-2009. This decline has continued, with the average intensity of the May-October subtropical ridge at 140-150°E increasing by 0.26 hPa/decade over the period 1979-2019. There has also been an increase in the mean sea level pressure and a weakening of the background zonal wind speeds in subtropical Australia. We find that the northern edge of the front and the likelihood of fronts producing rainfall show strong statistical relationships with the intensity and position of the subropical ridge, suggesting a possible link between trends in pressure and frontal rainfall. A link between decreasing frontal rain and increasing pressure was also identified by Burls et al. (2019) in southern Africa. However, the nature of this relationship depends on the location of the subtropical high and mechanisms may differ on frontal and non-frontal days. We also found that frontal rainfall is favoured during negative IOD conditions, consistent with Lawrence et al. (2022), and that IOD has a stronger influence on frontal rainfall than frequency. While robust trends cannot be calculated for the IOD over this short time period, paleoclimate data indicates that positive IOD events may be becoming more common (Abram et al., 2020).

An alternate explanation could be due to links between the mid-tropospheric frontal westerlies and the strength or location of the subtropical jet. In constrast to the observed strengthening of the subtropical ridge, there have been no robust trends identified in the southern hemisphere subtropical jet (Maher et al., 2019). However, a weakening of 700hPa zonal winds between 20-35°S during July was also identified for southeast and southwest Australia by Osbrough & Frederiksen (2021), which they linked to decreases in July baroclinicity and rainfall in southern Australia. Our analysis also indicates no change in the latitude of the subtropical jet (not shown), but a possible weakening in its intensity on front days in eastern Australia, although the average change in zonal wind speed is smaller at 300hPa than it is at 700hPa. The mean state changes in Figures 6 and S1 are also indicative of a modification of the ZW3 in the southern hemisphere , which has been associated with rainfall variability in parts of southeast Australia but has no clear trend in the cool season (Campitelli et al., 2021).

The strong link between frontal rainfall and changes in the subtropical 700hPa westerlies was tested for a second region that has experienced declines in cool season frontal rainfall, southwest Western Australia, with remarkably similar results. Easterly 300hPa wind anomalies and 850hPa moisture flux in the subtropics have also been noted during dry winters in South Africa (Mahlalela et al., 2019), which is another region which has experienced decreases in winter rainfall (Sousa et al., 2018; Burls et al., 2019) and weakening of the subtropical westerlies between 1980-1999 and 2000-2019. These results raise the question of what role changes in subtropical zonal winds may be playing in frontal rainfall in other areas of the southern hemisphere. Future work will develop a more generalisable approach to investigate how the northern extent of frontal westerlies is changing in other parts of the southern hemisphere, as well as how this is likely to change in the future climate given the projected poleward shift in the storm tracks and the edge of the Hadley Cell (IPCC, 2021). This may

provide a potential avenue to reconcile projected declines in cool season rainfall across southern hemisphere midlatitude regions (Lee et al., 2021) with the lack of change projected in frontal frequency (Catto et al., 2014).

**Data availability**

The merged front datasets for SEA and SWWA are available online at https://doi.org/10.6084/m9.figshare.20453325.v1. The front composites were developed using ERA5 data, which is freely available from the Copernicus Climate Change Service
Climate Data Store at https://cds.climate.copernicus.eu/#!/search?text=ERA5&type=dataset

**Acknowledgments**

This work has been supported by the Victorian Department of Environment, Land, Water and Planning through the Victorian Water and Climate Initiative, and has used computing resources from the National Computational Infrastructure. We thank Sugata Narsey, Pandora Hope, and two anonymous reviewers for their comments that have improved this manuscript.

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
