# Peer review of "Anomalous subtropical zonal winds drive decreases in southern Australian frontal rain"

_Weather and Climate Dynamics, 2022_

## Author Comment (AC1)

We thank both reviewers for their thoughtful comments on this manuscript. Our responses to the reviewer's comments are given below in bold, with the changes we will make in the edited manuscript given in italics.

**Response to Reviewer 1**

In this study the authors investigate the decrease in the accumulated rainfall in southeast Australia that is not a result of change in the frequency or intensity of the cold fronts. They analyze the properties of fronts in two twenty year periods and show that the change is due to a shift in the location of the maximum westerlies which can be a result of wind anomalies and changes in zonal wave 3.

I find that the presented results are interesting and offer a new explanation to the phenomena. Yet parts of the analysis need to be further explained in order to better convince the reader on the validity of the results. The changes, first in the total rain and then in the location of the winds, are all very small and a better job needs to be done to explain the importance of these small changes. There are many places in the text where the authors refer to non-significant increase/decrease seen in the figures, this can give the wrong impression and understanding of the results you do have.

The paper is well in the scope of WCD, however, I feel that some rewriting of the results in the text is required as well as addressing the questions below.

**Thank you for your detailed comments.**

While we agree that many of the changes shown are small, they can nevertheless have substantial downstream impacts – for instance, a relatively small ~11% decline in total annual rainfall during the Millennium Drought has previously been shown to result in a 46% decline in streamflow with significant socioeconomical impacts (e.g. https://doi.org/10.1002/wrcr.20123). Reduced streamflow has been associated with a change in the rainfall-runoff relationship, which unpublished work suggests may be partly a result in changes in the proportion of rainfall from different weather systems, especially fronts as they are particularly important for light and moderate rainfall during the cool half of the year.

**We will add more discussion of this to the paper, to better explain why the small changes we discuss are nevertheless important.**

The figures need to be improved, especially the median red line in most of the figures is not seen at all. In figure 2 (line 219), for example, there are markers for the statistically significant latitudes but the line itself shows no difference.

Thank you for this observation – we had some difficulties in PDF conversion and should have more thoroughly checked the figures. We have made the shading lighter and the lines thicker, so it is easier to see the median line. An example is below, but the full vector figures will be provided to the publication team with the revised manuscript.

Figure R1. A clearer version of Figure 2

Line 100: First, the sentence is not very clear. Is it comparable to panel 1d and its caption? Second, what is the reason to extend the front all the way to 20S? in the example in fig. 1 you extend the points to latitudes that there were no fronts detected in the first place. If you extend all the fronts in this way, does this extension that you are adding not affecting the results that you find regarding the shift in the latitude of maximum westerlies? How do you determine how far north to extend the front?

You are correct – we want to create composites relative to the front for the region 20-50S as a way of understanding the circulation both north and south of our domain of interest. The extension was only made to have a reference longitude for building composites for further analysis. In many cases, there is no front identified by the tracking scheme at these latitudes, but circulation at these latitudes may still be connected with the front and influence its rainfall – for instance, in many cases a trough or a "northwest cloudband" may extend from the northern edge into the tropics, increasing the moisture and rainfall in the front region. We will explain this in more detail, including referencing work on the interactions between fronts, troughs, cloudbands and atmospheric rivers for generating rain in this region (e.g. <a href="https://doi.org/10.1175/JCLI-D-21-0606.1">https://doi.org/10.1175/JCLI-D-21-0606.1</a>, <a href="https://doi.org/10.1175/JCLI-D-16-0686.1">https://doi.org/10.1175/JCLI-D-21-0606.1</a>, <a href="https://doi.org/10.1175/JCLI-D-16-0686.1">https://doi.org/10.1175/JCLI-D-16-0686.1</a>)

We will also rephrase the point to better match the caption:

**"Outside of the latitudes with an identified front, we infer an extended "front" longitude based on the last recorded front point, so that composites can be calculated over the full 20-50S region"**

Line 159-160: Fig.2 shows the total front rain, where the difference is not significant (according to the markings) and in the text you discuss the rain rate. I find this change in the accumulated rain very small, how does it compare to the differences in accumulated rain found in the other studies mentioned in the introduction?

The total front rain averaged across the region is mentioned in the next sentence ("a 32mm (9.4%) decline in total frontal rainfall in these latitudes between the two periods"); this change is not statistically significant (p=0.14), although it is comparable to other studies on declines in southeast

Australian frontal rainfall (e.g., a 35mm decline in https://doi.org/10.1002/joc.3597) and changes in total rainfall during the millennium drought (e.g., https://rmets.onlinelibrary.wiley.com/doi/abs/10.1002/joc.1627)

We will add to this section more discussion of the three time periods, as there is a statistically significant decline in total frontal rainfall between 1979-1996 and 1997-2009 (-14%, p=0.03), but a partial recovery in 2010-2019 (9% below the 1979-1996 average). We will also add an additional supplementary figure showing the three periods.

---

## Author Response (AR1)

We thank both reviewers for their thoughtful comments on this manuscript. Our responses to the reviewer's comments are given below in bold, with the changes we have made in the edited manuscript given in italics. Line numbers in all cases refer to the clean version of the manuscript, but a tracked changes manuscript as also been provided.

Response to Reviewer 1

In this study the authors investigate the decrease in the accumulated rainfall in southeast Australia that is not a result of change in the frequency or intensity of the cold fronts. They analyze the properties of fronts in two twenty year periods and show that the change is due to a shift in the location of the maximum westerlies which can be a result of wind anomalies and changes in zonal wave 3.

I find that the presented results are interesting and offer a new explanation to the phenomena. Yet parts of the analysis need to be further explained in order to better convince the reader on the validity of the results. The changes, first in the total rain and then in the location of the winds, are all very small and a better job needs to be done to explain the importance of these small changes. There are many places in the text where the authors refer to non-significant increase/decrease seen in the figures, this can give the wrong impression and understanding of the results you do have.

The paper is well in the scope of WCD, however, I feel that some rewriting of the results in the text is required as well as addressing the questions below.

**Thank you for your detailed comments.**

**While we agree that many of the changes shown are small, they can nevertheless have substantial downstream impacts – for instance, a relatively small ~11% decline in total annual rainfall during the Millennium Drought has previously been shown to result in a 46% decline in streamflow with significant socioeconomical impacts (e.g. https://doi.org/10.1002/wrcr.20123). Reduced streamflow has been associated with a change in the rainfall-runoff relationship, which unpublished work suggests may be partly a result in changes in the proportion of rainfall from different weather systems, especially fronts as they are particularly important for light and moderate rainfall during the cool half of the year.**

**We have added more discussion of why these small changes in rainfall are important to the conclusions at L448-457:**

*Frontal winds and rainfall have shown little recovery since the end of the Millenium Drought, and may be part of a longer-term decline in cool season rainfall over this region. While the total decrease in rainfall from fronts is relatively small, a 9.4% decline in frontal rainfall from 1980-1999 to 2000-2019 in southeast Australia, even small changes in rainfall can be magnified to much larger changes in streamflow and corresponding hydrological impacts. This was seen during the Millenium Drought, when a 11% decline in total annual rainfall was magnified to a 46% decline in streamflow in southeast Australia (Van Dijk et al., 2013). Changes in frontal rainfall may potentially play an important role in these declines, as fronts contribute strongly to the number of days with low to moderate rainfall intensity (Pepler et al., 2020) and the number of rain days is an important predictor of streamflow changes in southeast Australia (Fu et al., 2021). Consequently, any continuation of declines in frontal rainfall over coming decades may be of critical importance in catchments where runoff has yet to recover after the Millenium Drought (Peterson et al., 2021).*

The figures need to be improved, especially the median red line in most of the figures is not seen at all. In figure 2 (line 219), for example, there are markers for the statistically significant latitudes but the line itself shows no difference.

**Thank you for this observation – we had some difficulties in PDF conversion and should have more thoroughly checked the figures. We have made the shading lighter and the lines thicker, so it is easier to see the median line. All figures have been updated and an example is below. The full vector figures will be provided to the publication team with the revised manuscript.**

[Figure]

**Figure R1. A clearer version of Figure 2**

Line 100: First, the sentence is not very clear. Is it comparable to panel 1d and its caption? Second, what is the reason to extend the front all the way to 20S? in the example in fig. 1 you extend the points to latitudes that there were no fronts detected in the first place. If you extend all the fronts in this way, does this extension that you are adding not affecting the results that you find regarding the shift in the latitude of maximum westerlies? How do you determine how far north to extend the front?

**You are correct – we want to create composites relative to the front for the region 20-50S as a way of understanding the circulation both north and south of our domain of interest. The extension was only made to have a reference longitude for building composites for further analysis. In many cases, there is no front identified by the tracking scheme at these latitudes, but circulation at these latitudes may still be connected with the front and influence its rainfall. We have added additional explanation of the motivation for using this region for composites at L125-132:**

*While fronts can be as short as one degree in length, and the total number of identified fronts is lower at 20°S than at 50°S (Rudeva and Simmonds, 2015), we calculate composites for latitudes between 20-50°S for all fronts regardless of their latitudinal position within that interval. This is because in many cases an identified cold front can interact strongly with weather systems to the north of the identified front extent, such as troughs and northwest cloudbands (Reid et al., 2019, 2022), and impact the atmospheric circulation and rainfall patterns well into the tropics (Narsey et*

*al., 2017). Fronts are also often associated with atmospheric rivers that advect moisture from the tropics into higher latitudes (Reid et al., 2022), therefore understanding front-related circulation anomalies in the subtropics may help to understand changes in frontal rainfall.*

We also rephrased point 4 to better match the caption:

*"Outside of the latitudes with an identified front, we infer an extended "front" longitude based on the last recorded front point, so that composites can be calculated over the full 20-50°S region"*

Line 159-160: Fig. 2 shows the total front rain, where the difference is not significant (according to the markings) and in the text you discuss the rain rate. I find this change in the accumulated rain very small, how does it compare to the differences in accumulated rain found in the other studies mentioned in the introduction?

**The total front rain averaged across the region is mentioned in the next sentence ("a 32mm (9.4%) decline in total frontal rainfall in these latitudes between the two periods"); this change is not statistically significant (p=0.14), although it is comparable to other studies on declines in southeast Australian frontal rainfall. We have added additional discussion of the extent to which this rainfall trend is meaningful, including putting it into context of other studies and comparing frontal rainfall during the Millennium Drought and the pre- and post-drought periods at L190-197, and have provided a corresponding figure for the three periods below (Figure R2):**

*While this change is not statistically significant (p=0.14), it is similar in magnitude to the annual rainfall decline during the Millennium Drought (Van Dijk et al., 2013) and to declines in frontal rainfall reported in previous studies (Risbey et al., 2013b; Pepler et al., 2021).*

*There was a statistically significant decline in frontal rainfall between 1979-1996 and 1997-2009 (-14%, p=0.03), with a partial recovery during 2010-2019 (9% below the 1979-1996 period). The 2010-2019 period had very high variability, with very high frontal rainfall totals during 2010 and 2016 but low totals in other years, which explains the reduced significance of trends calculated using the later period.*

[Figure]

**Figure R2. As in Figure 2, but for three periods (1979-1996, 1997-2009 and 2010-2019), with crosses showing statistically significant changes between the first two periods.**

Line 220: The markings in figure 3 look between 28-33S, which one is correct?

**Thank you for checking this. Figure 3j does show that changes are stronger over the latitudes 28-33S; however, the text refers to changes in the latitudes 33-38S as those latitudes are more closely linked to frontal rainfall (see Figure 3d). To clarify this, we have changed the text at L257 to:**

*"There is also a statistically significant weakening of post-frontal relative vorticity over 33-38°S (-11%), with larger changes in vorticity over 28-32°S"*

Line 222: "While these changes are generally weak, and not necessarily aligned with the latitudes most relevant to our region of interest, together they are suggestive of an overall weakening of frontal baroclinicity" – this conclusion needs to be better explained.

**Thanks for this comment. We have replaced "overall weakening of frontal baroclinicity" with "weakening of uplift in the frontal area" demonstrated by weaking of pre-frontal vertical velocity, weak decrease in baroclinicity (to the south of our region of interest) and less subsidence behind the front, all of which is discussed in lines 255-260.**

Technical comments –

Line 46-47: this sentence is unclear and was hard to follow.

**We have rephrased this sentence to improve clarity (L44-49):**

*"The Millennium drought ended in 2009, and was followed by heavy rain during the subsequent La Niña years 2010-2011. However, while average annual rainfall over the 2010-2018 period was close to the long-term average (Fu et al. 2021) this recovery is predominantly associated with increased rainfall during the warm season, when a lower proportion of rainfall is converted into streamflow. In contrast, rainfall during the hydrologically important cool months of the year remained below the long-term average during the post-drought period 2010-2019 (Bureau of Meteorology and CSIRO, 2020; DELWP, 2020). "*

Line 69: "two points" would be 2 degrees? If the grid is 1 degree as stated in the next line

**We track fronts on a 1 degree grid and require that fronts are at least 2 grid points long. We have make this simpler in the text by saying "for this study we require that fronts are at least 2 grid points long and have at least one point in southeast Australia (30-40°S, 135-150°E)"**

Line 179: who has a more negative vertical velocity?

**Good catch; this sentence now says "Wet fronts have ..."**

Line 179: There is only one level presented, is this a result you found? If not please add reference.

**We performed our analysis over a range of levels, but only showed a single level in the figure as other levels are broadly similar. We have added more information on the levels used to L91-96:**

*Geopotential height (Z), horizontal (u, v) and vertical (w) velocity, relative vorticity, and temperature (T) were extracted and analysed on eight vertical levels (1000, 925, 850, 700, 600, 500, 300 and 200hPa). Most results are presented for 700hPa where changes were most significant, but results were broadly consistent across a range of levels. Single-level variables included mean sea level pressure (MSLP), rain rate, total column water (TCW), 500-1000 hPa vertically integrated moisture flux including its zonal and meridional components (IVT; Reid et al., 2022), and the Phillips Criterion (PC), a measure of baroclinicity.*

Line 191: should be figure 3f

**Fixed**

Line 219: maybe latitudes? And there is no (i) in fig. 2, please specify that it is in fig.3

**Fixed**

Line 331: STRI, STRP are not defined. So are some of the other indexes that appear in the following lines. IOD is defined only on line 356 but used before.

**All indices should now be defined at L148-156, including STRI, STRP, and IOD:**

*Pearson's correlation coefficients are calculated using linearly detrended data to assess relationships between seasonal mean frontal characteristics and the intensity (STRI) and position (STRP) of the subtropical ridge calculated over 140-150°E (Timbal and Drosdowsky, 2013). We also calculate correlations with the Troup (1965) Southern Oscillation Index (SOI; http://www.bom.gov.au/climate/enso/soi/), an indicator of the El Niño-Southern Oscillation (ENSO); the Dipole Mode Index (DMI), an indicator of the Indian Ocean Dipole (IOD: Saji et al., 1999; https://stateoftheocean.osmc.noaa.gov/sur/ind/dmi.php); and the Southern Annular Mode (SAM: https://www.cpc.ncep.noaa.gov/products/precip/CWlink/daily_ao_index/aao/aao.shtml).*

Table 1 title: Pearson's correlations

**Thanks, fixed**

Line 367: missing reference?

**We have added additional clarification at L412:**

**"Similarly, while there was a -0.4°/decade linear trend in the average northernmost latitude with U700>=10m/s (p=0.01), removing variability associated with STRI decreased this trend to -0.2°/decade (p=0.06)."**

Figure A2: it is like figure 5, not 4

**Thanks, fixed**

**Overview:**

This study investigates the possible causes behind the decline in frontal rainfall in southeast Australia (SEA). It builds on previous regional studies that have highlighted that there is little evidence to suggest that frequency of fronts have not changed, yet there has still been a decline in rainfall. Using a wind-based front detection method, the authors investigate what frontal characteristics could be behind the decline in rainfall within SEA. The key result being that decline in rainfall in SEA is linked to a weakening and southward shift in the northern edge of the front. Although there are numerous complex factors that impact the northern extent of the fronts, the intensity and position of the subtropical ridge appears to play a key role.

Overall, the study contains interesting scientific results, has appeal to a wide audience and is well presented. Thus, it should be considered for publication after some revision.

**Thank you for your helpful comments**

**Major Comments:**

Perhaps one of the key findings here is that there is evidence that the northernmost latitude at which fronts extend into the region has shifted polewards (i.e. weaker zonal winds). However, it is not clear as to how this results in less rainfall. It is suggested that it could be measure of frontal intensity, but perhaps just some more discussion as to the why or how is needed. Could it be less moisture transported into the region or perhaps less moisture already in the region before the front passes through?

**This is a good question. We have added some additional discussion of possible mechanisms based on wind and moisture transport at L459-471, before the discussion of more complex mechanisms related to changes in the subtropical ride and subtropical jet.**

***The link identified in this paper between subtropical wind changes and changes in frontal rainfall is a statistical link, and would require further modelling studies to understand the dynamical interaction between subtropical winds and rainfall ~10° further south. However, there are a number of potential mechanisms. The southward shift in frontal westerlies could simply be an indication of a southward shift in the latitude at which fronts occur, which would suggest that the front detection method based on wind changes is deficient at detecting the equatorward extent of cold fronts and subtle changes at these latitudes. However, front detection methods based on other variables such as temperature gradients also do not identify any changes in frontal frequency (Pepler et al., 2021; Berry et al., 2011). Another potential mechanism is a link between weakening subtropical westerlies and a reduction in atmospheric moisture flux into the region; while there has been little change in total column water or IVT in SEA, both TCW and IVT have decreased in the region 28-33°S associated with the weakening subtropical winds. Finally, while we found only a weak change in vertical velocity and no significant change in PC, changes in other factors such as convective inhibition or the level of condensation in a warming climate could also reduce the ability of fronts to generate rainfall, particularly during the cool season and moderate rain events where convection is weak (Rasmussen et al., 2020).***

**We also looked more closely at the link between moisture and frontal rainfall – while there has been no real change in e.g. IVT in the region 33-38S (average of 203.9 in 1980-1999 vs 201.8 in 2000-2019), there has been a decline in IVT further north at 28-33S (161.5 in 1980-1999 to 152.2 in 2000-2019 (p<0.001)), mostly due to changes in the zonal component (Figure R3). While backtracking trajectories (e.g., Holgate 2022) generally suggest moisture for fronts in this region is**

sourced from the ocean to the southeast, some moisture is still advected from the ocean to the north-east, so weakening winds to the north of the region could reflect an overall decrease in moisture flux into the region, as there is a correlation of 0.63 between frontal rain and IVT in the region 28-33S. We have added a brief mention of this relationship to the text at L267:

*Associated with the weakening zonal winds, there has also been a reduction in mean IVT at 28-33°S (not shown), indicating a reduction of moisture flux at the northern edge of the front, although there has been no change in IVT over SEA (33-38°S).*

[Figure]

**Figure R3 – As in Figure 3, but for IVT (left) and its zonal (b) and meridional (c) components on wet vs dry fronts (left) and 1980-1999 vs 2000-2019 (right)**

**Minor Comments:**

A domain map early on in the manuscript would help the reader understand the domain of the study. This could be used to showcase the domains described in lines 70-71. A satellite image of day used to describe the front detection method could be an example of a domain map.

**Thanks, this is a good suggestion! While satellite data is a good idea, we thought it would be more useful for the domain map to show an example of key fields from ERA5 on a given day – MSLP, rain, and the location of identified fronts. We have added the figure below to the manuscript as a new Figure 1.**

[Figure]

**Figure R4. ERA5 rain rate (shading, mm/h), MSLP (contours) and all identified cold fronts (red lines) for 600UTC on 9 June 2004. The SEA (orange) and SWWA (purple) regions used for front identification are also marked.**

Consistency between the use of 'Southern Hemisphere' (e.g. line 7 and line 24) and 'southern hemisphere' (e.g. line 18 and 21) throughout the manuscript.

**Thanks, we have changed all mentions to lower case**

Line 84 – Is there any evaluation on ERA5 precipitation data with that of stations in the region? Not necessary for this study, just out of general interest for the reader in terms of decline seen in ERA5 compared to that in the observations.

**This is a good question, and we have added some additional discussion of this to the methods section at L85-89:**

*While reanalyses often evaluate poorly against observed rainfall measurements (Alexander et al., 2020), ERA5 generally evaluated well over Australia (Lavers et al., 2022). The predecessor to ERA5 (ERA-Interim) was found to generally perform well in simulating frontal rainfall and moderate rainfall intensities over the oceans near Australia (Lang et al., 2018), despite deficiencies in simulating pre-frontal and non-frontal rainfall, making ERA5 well suited to this study.*

Line 92-100 – Consider linking each bullet point to the associated figure panel in Fig. 1.

 **Good idea, we've done that.**

Line 100 – Could you explain Point 4 in more detail. It is not clear as to why this has been done.

**We want to create composites for the region 20-50S as a way of understanding the circulation both north and south of our domain of interest at the time when a front is found in our region of interest. We have explained this in more detail at L125-132:**

*While fronts can be as short as one degree in length, and the total number of identified fronts is lower at 20°S than at 50°S (Rudeva and Simmonds, 2015), we calculate composites for latitudes between 20-50°S for all fronts regardless of their latitudinal position within that interval. This is because in many cases an identified cold front can interact strongly with weather systems to the north of the identified front extent, such as troughs and northwest cloudbands (Reid et al., 2019, 2022), and impact the atmospheric circulation and rainfall patterns well into the tropics (Narsey et al., 2017). Fronts are also often associated with atmospheric rivers that advect moisture from the tropics into higher latitudes (Reid et al., 2022), therefore understanding front-related circulation anomalies in the subtropics may help to understand changes in frontal rainfall.*

Line 149 – Colorbar needs a label and units

 **Units are in mm/h, which is listed in the caption**

Line 160 – 32mm per season?

**We have corrected this to "32mm (9.4%) decline in the average frontal rainfall per season"**

Line 171 – Typo - 'South African'

**Fixed**

Line 186/187 – I can understand why this is done, but perhaps for consistency in the Phillips Criterion acronym (PC) can just be used as it has already been used earlier.

**Ok, changed to PC**

Line 190 – does this infer that wet fronts move more slowly through the region compared to dry fronts?

**This is an interesting question, which is difficult to assess as individual fronts are not tracked between time steps.**

**One thing we tried is looking at the number of consecutive time steps with a front identified in the region, although this does not distinguish between cases of a single slow-moving front or multiple consecutive fast-moving fronts. We found that front "events" of less than 24 hours had lower average rain rates than longer events (0.09 mm/h vs 0.16 mm/h), but there was no change between 1980-1999 and 2000-2019 in the average duration of front conditions (20h vs 19.4h) or in the number of front "events" of 24 hours or more (35.45 vs 35.10).**

Line 201 – Should the label for the blue in Fig 3. a-f not be "wet" instead of Rain>=0.1mm/hr? That would just keep it consistent with the text. You could also consider splitting Fig. 3 into two separate figures.

**We prefer to keep Figure 3 as a single figure as it makes it easier to compare whether the latitudes relevant to frontal rain and the latitudes with significant changes align. But we have increased the font size slightly and changed the legend to say wet, see figure R1 in our response to Reviewer 1**

Line 201 – Could the font not be made larger in Fig. 3? The mean lines are also difficult to read in the figure.

**We have made the shading lighter and the lines larger in figure 3 for ease of comparison, see figure R1 in our response to Reviewer 1**

Line 251 – Similar to an earlier comment. Can one infer that the speed at which a front passes through a domain has a direct effect on the volume of rainfall produced?

**This is discussed briefly in response to your previous comment – while this seems intuitively reasonable it is difficult to assess using our datasets**

Line 276 – are the results for this section any different if the winter is broken down into an early vs late winter?

**This is a good question. Monthly data has larger variability, which means changes in individual months or shorter periods are generally not statistically significant, which is why we focused on the extended period in the paper.**

**When comparing 1979-1996 and 1997-2009 (Figure R5a), there are clear declines in frontal rainfall for all months April-October. However, in the period 2010-2019 frontal rainfall in the early season April-June has mostly recovered, but July-October has not. This is now briefly mentioned at L197:**

*Recovery during the later period occurred mostly in the early season, April-June, with the frontal rainfall anomaly in July-October during 2010-2019 (-15% compared to July-October 1979-1996) similar to that in 1997-2009 (-17%).*

**Looking at days with strong frontal westerlies in the subtropics (Figure R5b), these are generally only seen for fronts during the cool months, and have declined in frequency for all months (except May)**

[Figure]

**Figure R5. a) Total accumulated frontal rainfall by month for the three periods of interest. B) Number of front days per month where U700>=10 m/s is recorded at least as far north as 23°S.**

Line 285 – Consider re-ordering the panels in Figure 5.

**Thanks, good idea. We have reordered the panels in the final manuscript to place the mean rainfall first (a), as it is the first panel discussed in the text.**

Line 325 – Is there any intraseasonal variation with the relationship between the different drivers and frontal frequency / rainfall? For example, does the IOD have a stronger relationship with frontal rainfall during the late winter months compared to the early winter months?

**This is another good question. While DMI and STR are significantly correlated with the frontal rainfall and the latitude of the frontal westerlies in both May-July and August-October (Table R1),**

the DMI link strengthens considerably in the second part of the season as this is when the DMI gets stronger. We have added text to that effect in the text at L395-398:

*"Given that the IOD is most active in spring, the associated between IOD and Australian rainfall is strongest around the springtime (e.g., Cai et al. 2011). The correlation between the DMI and total frontal rainfall also strengthens from -0.39 during May-July to -0.65 during August-October. In both seasons negative IOD is associated with an increased frequency of wet fronts and a northward shift in latitudes with frontal westerlies, but no significant change in the total number of fronts"*

**Table R1. As in Table 1, but for May-July and August-October separately. Bold indicates significance for $p<0.05$.**

| Variable | STRI | STRP | SOI | NINO3.4 | DMI | SAM | NINO3.4 - DMI | DMI - NINO3.4 |
|---|---|---|---|---|---|---|---|---|
| Front days MJJ | -0.13 | **-0.42** | 0.04 | 0.02 | 0.09 | 0.01 | -0.02 | 0.10 |
| Total front rainfall MJJ | **-0.64** | **-0.50** | **0.50** | -0.28 | **-0.39** | -0.06 | -0.15 | -0.31 |
| Number of wet fronts MJJ | **-0.54** | **-0.60** | 0.29 | -0.19 | -0.23 | -0.10 | -0.12 | -0.19 |
| Average latitude of U>=10m/s MJJ | **-0.64** | **-0.50** | **0.49** | -0.18 | **-0.43** | -0.05 | -0.01 | **-0.41** |
| Front days ASO | **-0.65** | **-0.73** | 0.35 | -0.25 | -0.28 | -0.34 | -0.14 | -0.16 |
| Total front rainfall ASO | **-0.61** | **-0.48** | 0.44 | **-0.38** | **-0.65** | 0.19 | -0.04 | **-0.60** |
| Number of wet fronts ASO | **-0.74** | **-0.62** | 0.48 | **-0.42** | **-0.65** | 0.09 | -0.10 | **-0.58** |
| Average latitude of U>=10m/s ASO | **-0.84** | **-0.70** | 0.45 | -0.32 | **-0.53** | 0.10 | -0.02 | **-0.49** |

Line 339 – Typo - 'Pearson's

**Thanks, fixed**

Line 340 – Typo - 'm/s'

Fixed

Line 348 – Could this be linked to anomalous moisture transport from the tropical Indian Ocean? Or is the impact more of changes on the Hadley Circulation and a weaker / stronger STR?

**Most of the moisture for rainfall events in southeast Australia is sourced from the southern ocean as well as from the ocean to the east (e.g. https://doi.org/10.1175/JCLI-D-19-0926.1 supplementary figure S11b, "southeast coast Victoria") and previous work by, e.g., Cai suggests that the IOD mostly influences Australia's climate via a wave train in the extratropics (https://doi.org/10.1175/2011JCLI4129.1). We have briefly addressed this at L392:**

*While the link between IOD and frontal rainfall could in part be associated with increases in moisture from the tropics, the IOD was also associated with a shift in the average northernmost latitude of strong frontal westerlies, consistent with the extratropical pathway of IOD impacts (Cai*

*et al., 2011) and the link between the IOD and the mean westerlies in southeast Australia (Pepler et al., 2014).*

Line 368 – Just a general comment, what role does topography and wind direction play? Could this account for any minor differences between the two Australian domain?

**Southwest Western Australia and southeast Australia have very similar mean climates, so are often clustered together (e.g., in https://doi.org/10.1071/ES20003). However, you are correct that there are differences in regional topography and wind interactions, as well as other factors such as ocean temperatures and coastal orientation, that can contribute to differences between the regions. For example, the regions typically source moisture for rainfall from different ocean basins and have different relationships with climate drivers such as ENSO and IOD. Given these differences, the similarities between the two regions regarding frontal rainfall in this paper were stronger than expected, although both regions have seen related declines in cool season rainfall (e.g., https://doi.org/10.1002/joc.1964).**

Line 427 - 'South Africa'
**Thanks, fixed**